# Study of the Influence of the Reprocessing Cycles on the Final Properties of Polylactide Pieces Obtained by Injection Molding

**DOI:** 10.3390/polym11121908

**Published:** 2019-11-20

**Authors:** Angel Agüero, Maria del Carmen Morcillo, Luis Quiles-Carrillo, Rafael Balart, Teodomiro Boronat, Diego Lascano, Sergio Torres-Giner, Octavio Fenollar

**Affiliations:** 1Technological Institute of Materials (ITM), Universitat Politècnica de València (UPV), Plaza Ferrándiz y Carbonell 1, 03801 Alcoy, Spain; anagrod@epsa.upv.es (A.A.); mamores@epsa.upv.es (M.d.C.M.); luiquic1@epsa.upv.es (L.Q.-C.); rbalart@mcm.upv.es (R.B.); tboronat@dimm.upv.es (T.B.); ocfegi@epsa.upv.es (O.F.); 2Escuela Politécnica Nacional, Quito 17-01-2759, Ecuador; 3Novel Materials and Nanotechnology Group, Institute of Agrochemistry and Food Technology (IATA), Spanish National Research Council (CSIC), Calle Catedrático Agustín Escardino Benlloch 7, 46980 Paterna, Spain; storresginer@iata.csic.es

**Keywords:** PLA, extrusion cycles, injection molding, mechanical recycling, circular economy

## Abstract

This research work aims to study the influence of the reprocessing cycles on the mechanical, thermal, and thermomechanical properties of polylactide (PLA). To this end, PLA was subjected to as many as six extrusion cycles and the resultant pellets were shaped into pieces by injection molding. Mechanical characterization revealed that the PLA pieces presented relatively similar properties up to the third reprocessing cycle, whereas further cycles induced an intense reduction in ductility and toughness. The effect of the reprocessing cycles was also studied by the changes in the melt fluidity, which showed a significant increase after four reprocessing cycles. An increase in the bio-polyester chain mobility was also attained with the number of the reprocessing cycles that subsequently favored an increase in crystallinity of PLA. A visual inspection indicated that PLA developed certain yellowing and the pieces also became less transparent with the increasing number of reprocessing cycles. Therefore, the obtained results showed that PLA suffers a slight degradation after one or two reprocessing cycles whereas performance impairment becomes more evident above the fourth reprocessing cycle. This finding suggests that the mechanical recycling of PLA for up to three cycles of extrusion and subsequent injection molding is technically feasible.

## 1. Introduction

Biopolymers can be defined as polymers that are not “harmful” for the environment. The concept of a biopolymer is strongly linked to the origin of the material (petrochemical or natural resource) and to its end-of-life (whether it is biodegradable or not) [1,2,3,4]. Biopolymers include petroleum-derived polymers with the biodegradation (disintegration in controlled compost soil conditions) feature. Many of these polymers have ester groups that can be easily hydrolyzed in the presence of moisture [5]. Some aliphatic and aromatic polyesters such as poly(ε-caprolactone) (PCL), poly(butylene succinate) (PBS), poly(glycolic acid) (PGA), poly(butylene adipate-*co*-terephthalate) (PBAT), among others, are included in this group [6,7,8,9]. On the other hand, some research studies have been focused on obtaining conventional polymers from natural resources, but these are not biodegradable as their corresponding counterparts. Their properties are almost identical to their respective petrochemicals but are obtained by a renewable route. It is worthy to note the growing use of bio-based high-density polyethylenes (bio-HDPE) obtained from sugarcane [10,11,12]. Bio-polyamides (bio-PAs), such as polyamide 610 (PA610) or polyamide 1010 (PA1010), are also partially or totally obtained from renewable resources [13,14]. However, the most environmentally promising biopolymer group includes biopolymers from renewable origin and are biodegradable. This group is divided into three large subgroups: (a) polysaccharides and derivatives such as starch, cellulose, chitosan, chitin, and so on. This group also includes polylactide (PLA) in which the monomer is obtained by fermentation of starch-rich compounds [15,16], (b) Proteins such as gluten, soy protein, collagen, keratin, among others, which are used in the manufacturing materials for engineering or bioengineering [17,18,19], and (c) Bacterial polymers such as polyhydroxyalkanoates (PHAs), where some bacteria can synthetize up to more than 300 of these polyhydroxy acids obtained from butyric acid or valeric acid, among others [20,21]. Biodegradable polymers are defined by their unique disintegration capacity under controlled compost conditions (humidity, temperature, microbial strains, etc.). During the biodegradation, the polymer undergoes a series of processes that lead to the progressive breakage of long polymer chains into shorter segments that can enter into the life cycle of certain microorganisms [5,6,22,23]. One interesting group of biodegradable polymers is that of polyesters (mainly aliphatic polyester from petroleum or natural resources) since the ester group is susceptible to hydrolysis in the presence of moisture [5,7,24].

PLA has established itself as one of the current biopolymers with the most significant potential in various industrial areas [25,26]. PLA is currently one of the world’s most consumed biopolymers [27]. By copolymerization, materials with tailored properties can also be obtained. For instance, copolymers of lactic acid with glycolic acid (PLA-*co*-PGA) and with caprolactone (PLA-*co*-PCL) can find uses in engineering applications [28,29,30]. Since PLA and PLA-based copolymers are also resorbable, it is possible to find biomedical grades of these materials with increasing uses in biotechnology [31,32]. In the biomedical field, therefore, the use of PLA and its copolymers are continuously finding novel applications such as surgical sutures, bone implants, tissue, and prosthetic implants, fixing plates, fixing screws, porous scaffolds for cell growth, etc. [33,34,35]. At the same time, pharmaceutical companies are also developing PLA microcapsules and nano-capsules for the controlled release of drugs [15,36]. Due to the excellent biodegradability and biocompatibility properties, PLA applications have been growing at the industrial level in other areas such as textile, packaging, automotive, and the construction sector. In addition, with the recent emergence of additive manufacturing technologies based on 3D printing, PLA is one of the widely used materials in this industry [37,38,39,40]. In the packaging sector, PLA can be used for liquid containers, glasses, plates, disposable cutlery, frozen food packaging, and clear food packaging [41,42,43].

Due to the increasing use of PLA, along with biodegradation, the possibility of mechanical (secondary) recycling has also been considered [44]. However, due to the high sensitivity to hydrolysis of its ester groups at the processing conditions, recycled PLA materials have lower properties compared to the virgin material. Hydrolysis is responsible for a decrease in the polymer chain length, an increase in crystallinity, a reduction in toughness, biodegradation acceleration, and more [45,46,47,48]. As a result, some research works have been focused on the use of different additives to minimize this effect by preventing PLA chains from hydrolysis [49,50]. Another approach is focused on increasing the polymer chain length after hydrolysis by using chain extenders [44,51,52,53,54]. Chain extenders also play a crucial role in processing and then recycling since they can contribute to maintaining good melt strength and other rheological properties during manufacturing [55,56]. In some cases, not only chain extension can help to improve the features of a partially hydrolyzed PLA but also some cross-linking after γ irradiation can be attained [57]. Some authors have reported the role of poly(L-lactide) (PLLA) and poly(D-lactide) (PDLA) isomers to minimize the sensitiveness to hydrolysis of PLA materials. For instance, Karst et al. reported that stereo-complex PLLA/PDLA blends at 50/50 (wt/wt) show the highest resistance to hydrolysis [46]. By taking into account the relevance of the hydrolysis process and its role in the potential PLA recycling, the main goal of this research work is to evaluate the influence of reprocessing cycles carried out by extrusion on injection-molded pieces of PLA in terms of their mechanical, thermal, and thermomechanical properties, and changes in morphology, visual appearance, and melt behavior.

## 2. Materials and Methods

### 2.1. Materials

PLA commercial-grade Ingeo^TM^ 2003D, supplied by Natureworks (Minnetonka, MN, USA), was used to evaluate the effect of the reprocessing cycles. This PLA grade has a density of 1.24 g·cm^−3^ and a melt flow index (MFI) of 6 g/10 min (measured at 210 °C and with a load of 2.16 kg). This PLA grade offers high transparency and finds applications in the food-packaging sector.

### 2.2. Processing and Reprocessing of PLA

Before processing and reprocessing, PLA was subjected to a drying process at 60 °C for 4 h in an air circulating oven to remove the residual moisture. This moisture can be harmful during the processing at elevated temperatures due to the sensitiveness of ester groups to hydrolysis.

The processing and subsequent reprocessing cycles of PLA were carried out in a twin-screw co-rotating extruder with a screw diameter of 30 mm, supplied by Construcciones Mecanicas Dupra, S.L. (Alicante, Spain). The temperature profile used was 180 °C (feeding hopper), 185 °C, 190 °C, and 195 °C (extrusion die). The rotating speed remained at around 22 rpm. After extrusion, the strands were cooled in air and then pelletized using an air-knife unit. One part of the pellets was separated for shaping by injection molding to obtain standard samples and the remaining material was subjected to another processing cycle in identical conditions. The same material was reprocessed up to six times. The resultant samples were labeled as “PLAi,” where i = 0 to 6, and where 0 represents the raw material directly processed by injection molding without any extrusion cycle.

Each of the compounded pellets of the PLAi samples were dried and then subjected to injection molding in a Meteor 270/75 from Mateu & Solé (Barcelona, Spain). The temperature profile was 160 °C (hopper), 165 °C, 170 °C, and 175 °C (injection nozzle). The injection time was kept at 1 s and the cooling time was 10 s. These processing conditions were set for all samples. Then, two types of samples were obtained. A dog-bone shaped specimen with the dimensions recommended by ISO 527-4 and 4-mm thickness and a rectangular specimen with dimensions 80 × 10 × 4 mm^2^. These geometries are standard shapes for a wide variety of characterization techniques.

### 2.3. Mechanical Characterization

Tensile tests were carried out using a universal testing machine ELIB 30 from S.A.E. Ibertest (Madrid, Spain), according to ISO 527-4. A 5-kN load cell was used, and the cross-head rate was set to 5 mm·min^−1^. The tensile strength (σ_t_) and the percentage of elongation at break (%ε_b_) were obtained from the corresponding stress-strain diagrams. At least six different specimens were tested. To obtain accurate values of the tensile modulus (E_t_), an axial extensometer from S.A.E. Ibertest model IB/MFQ-R2 (Madrid, Spain) was coupled to the tested samples using a cross-head speed of 2 mm·min^−1^ and the corresponding values of the tensile modulus were measured and averaged. The impact resistance is directly related to toughness and, therefore, the energy absorption capacity before failure occurs. A Charpy pendulum from Metrotec S.A. (San Sebastian, Spain) was used for this test, with an energy of 6 J, according to the indications of ISO 179. Five different unnotched specimens were tested to obtain reliable values. Hardness values were collected using a Shore durometer model 673-D by J. Bot Instruments (Barcelona, Spain), according to ISO 868:2003. Shore D values were measured after a stabilization time of 10 s. To obtain reproducible values, at least 10 measurements on five different samples were taken in different areas. All mechanical tests were performed at room conditions.

### 2.4. Morphological Characterization

The morphology of the fracture surfaces after the impact tests was studied in a field emission scanning electron microscope (FESEM) supplied by Oxford Instruments (Abingdon, UK), working with an electron acceleration voltage of 1.5 kV. Prior to observation, the samples underwent a sputtering coating in a SC7620 sputter-coater supplied by Quorum Technologies Ltd. EMITECH (East Sussex, UK) to be conductive. The coating was carried out under high vacuum conditions using a gold-palladium alloy target for the sputtering in an argon atmosphere.

### 2.5. Characterization of the Melt Flow Index

The determination of the melt flow index (MFI) was carried out according to ISO 1133 with an MFI equipment from Metrotec S.A. (San Sebastian, Spain) using a 1-mm diameter nozzle at a temperature of 210 °C and a mass of 2.16 kg. The elapsed time between two consecutive measurements was 15 s. The degree of degradation for reprocessed PLA was estimated based on the variation of MFI by calculating the polymer degradation index (PDI), as suggested by Rojas-González et al. [58]. PDI represents the degradation rate by the effect of the reprocessing cycles and is a quantitative measure of the processing stability of a polymer material [59]. This parameter was estimated for each PLAi sample according to the following expression.
(1)$$PDi=abs\left[\frac{\left(MFI_{0}+MFI_{1}+\ldots +MFI_{n}\right)-\left(n-1\right)MFI_{0}}{\left(n+1\right)MFI_{0}}\right]$$
where *abs* means absolute value, *n* is the number of reprocessing cycles, and MFI_i_ is the melt flow index corresponding to PLAi after *i* extrusion cycles (for virgin polymer *i* is 0). Degradation is proportional to PDI. Briefly, an ideal, non-degraded polymer has a PDI of 0, while, when PDI approaches 1, it is representative of very high degradation.

### 2.6. Thermal Characterization

Analysis using differential scanning calorimetry (DSC) provides an evident view of the thermal transitions of polymer materials. The guidelines of UNE EN ISO 11357 were used to evaluate the effect of the reprocessing cycles on the thermal properties of PLA. A DSC equipment from Mettler-Toledo S.A., model DSC821e (Barcelona, Spain), was used. The thermal program selected for this study consisted of three cycles: two heating and one cooling steps. The first step consisted of a dynamic heating from 30 °C to 200 °C. The purpose of this first heating was to remove the thermal history of the material. The second step of the thermal cycle was a cooling ramp scheduled from 200 °C down to 0 °C. Lastly, a second heating stage was carried out from 0 °C up to 350 °C. All the heating/cooling rates were set to 10 °C /min in a nitrogen atmosphere with a flow-rate of 66 mL·min^−1^. From the second heating DSC thermogram, the most relevant thermal properties of PLA and reprocessed PLA (PLAi) were obtained: glass transition temperature (*T_g_*), enthalpy and cold crystallization temperature *(ΔH_cc_* and *T_cc_*), and enthalpy and peak melting temperature (Δ*H_m_* and *T_m_*). From the area involved by the melting peak, the maximum crystallinity (χ_c_max_) that the PLA acquires as the reprocessing cycles increase was estimated, according to the following expression.
(2)$$\%\chi_{c\_max}=\frac{\Delta H_{m}}{\Delta H_{m}^{0}}\cdot 100$$
where Δ*H_m_^0^* represents the theoretical melting enthalpy for a fully crystalline PLA, which was assumed to be 93 J g^−1^ [60].

To evaluate the thermal stability, a thermogravimetric analysis (TGA) was carried out using a TGA1000 from Linseis (Selb, Germany). All samples were subjected to a dynamic heating program from 30 to 700 °C at a constant heating rate of 20 °C·min^−1^ under air atmosphere. All the sample’s weight rate was between 10–20 mg, which maintained the same proportion in all samples to compare the results. Standard alumina crucibles of 70 μL were used. The main thermal parameters from TGA were the onset degradation temperature, which was assumed at a weight loss of 5 wt% (*T_5%_*), and the maximum degradation rate temperature (*T_deg_*).

### 2.7. Thermomechanical Characterization

Dimensional stability was evaluated using thermomechanical analysis (TMA) on squared samples of 10 × 10 mm^2^ with an average thickness of 4 mm. The study was carried out in a Q400 thermo-analyzer from TA Instruments (New Castle, DE, USA). A temperature sweep from 0 °C up to 140 °C was scheduled, with a heating rate of 2 °C·min^−1^ and an applied normal force of 20 mN. From the slope of the change in dimension vs. temperature graphs, the linear thermal expansion coefficients (CLTE) below and above the *T*_g_ were calculated.

Thermomechanical properties were determined by a dynamic mechanical thermal characterization. It was performed by an AR-G2 oscillatory rheometer from TA Instruments (New Castle, DE, USA), which was equipped with a special clamp made for solid samples (10 × 40 × 4 mm^3^) and works in torsion and shear conditions. Specimens were subjected to a temperature sweep from 30 °C to 140 °C at a heating rate of 2 °C·min^−1^, whereas the frequency and the maximum shear deformation were set at 1 Hz and 0.1%, respectively. With this test, it was intended to analyze the viscoelastic behavior of the PLA materials as a function of temperature, time, and/or frequency of the dynamic stress. The test consisted basically of the application of a dynamic sinusoidal stress model that allows the determination of the mechanical properties from the gap between the applied stimulus (tension) and the response of the material (deformation). The values of the storage modulus (*G*′) and dynamic damping factor (tan *δ*) were obtained. All thermomechanical tests were done in triplicate.

### 2.8. Color Measurements

The influence of the reprocessing cycles on the color of the PLA pieces was carried out in a Konica CM-3600d COlorflex-DIFF2, from Hunter Associates Laboratory, Inc. (Reston, VA, USA). The color coordinates *L**, *a**, and *b** were collected. *L** represents the luminance (black to white), *a** indicates the change between green and red, and *b** represents the change from blue to yellow. The colorimeter was calibrated with a white standard tile and a mirror device for the black (no light reflection). The white color standard with coordinate values *L*a*b** were obtained on five different samples and the corresponding average values and standard deviations were calculated. From this coordinate, it was possible to determine the color difference associated with this space. The distance metric, Δ*E*_ab_*, was obtained by following Equation (3) and compared with the color coordinates of the neat PLA (PLA0) piece. By following Equation (4), the yellowness index (YI) was obtained, where X, Y, and Z are the CIE color space values. To this end, the ASTM E313 method was used by considering an illuminant D65 and a standard observer function of 10°.

(3)$$\Delta E_{ab}^{*}=\sqrt{{\left(\Delta L^{*}\right)}^{2}+{\left(\Delta a^{*}\right)}^{2}+{\left(\Delta b^{*}\right)}^{2}}$$

(4)$$YI=\frac{100\left(1.3013\;X-1.1498\;Z\right)}{Y}$$

Color change was evaluated with the following assessment: Unnoticeable (ΔE*_ab_  <  1), only an experienced observer can notice the difference (ΔE*_ab_  ≥  1 and < 2), an unexperienced observer notices the difference (ΔE*_ab_  ≥  2 and < 3.5), clear noticeable difference (ΔE*_ab_  ≥  3.5 and < 5), and the observer notices different colors (ΔE*_ab_  ≥  5) [61,62].

### 2.9. Statistical Analysis

The mechanical and thermal properties were evaluated through analysis of variance (ANOVA) using STATGRAPHICS Centurion XVI v 16.1.03 from StatPoint Technologies, Inc. (Warrenton, VA, USA). Fisher’s least significant difference (LSD) was used at the 95% confidence level (*p* < 0.05). Mean values and standard deviations were also calculated.

## 3. Results

### 3.1. Influence of the Reprocessing Cycles on the Mechanical and Morphological Properties of PLA

PLA is a highly sensitive polymer during processing. Even a single processing cycle could potentially cause deterioration in the material because of its sensitivity to hydrolysis, which leads to a reduction in molecular weight (*M*_W_) and, consequently, promotes changes in other properties linked to the polymer structure [52,63]. These include mechanical properties. It is clear that if a single processing cycle can affect the features of the PLA, the increasing number of reprocessing cycles will have, presumably, a significant impact on the overall properties of the PLA. This could be a critical factor in its potential mechanical recycling in a similar way as it happens with polyethylene terephthalate (PET) recycling [64].

Table 1 shows a summary of the most relevant parameters obtained by tensile tests. The PLA pieces processed without any extrusion cycle (PLA0) showed a σ_b_ value of 54.8 MPa, together with a E_t_ value of 3580 MPa, which indicates that it is a rigid material with a relatively high tensile strength and modulus, comparable to that of polystyrene (PS) and PET. The E_t_ value was accurately obtained by using an axial extensometer, which allows accurate calculus of the applied stress and elongation and, subsequently, more precise values of the tensile modulus can be obtained [65,66]. However, ε_b_ was noticeably low, which indicates that it is a brittle material. The values of ε_b_ for the PLA0 pieces was only 9.73%, which is why many research studies have been focused on increasing PLA toughness with different approaches (blending, plasticization, chain extenders, etc.) [67]. As can be seen in Table 1, the number of reprocessing cycles significantly influenced the tensile properties. As indicated previously, hydrolysis can promote chain scission. As the polymer chains become shorter, they are more readily to pack in an ordered structure, which leads to increased crystallinity. The effect of increased crystallinity on the mechanical properties can also increase the mechanical strength of the PLA pieces. This increase observed in the E_t_ was directly related to other mechanical properties since this property can be defined as the stress-to-strain ratio in the linear deformation region in a typical stress-strain diagram. The values of ε_b_ progressively decreased down to values of around 6.28% in the PLA pieces subjected to six reprocessing cycles. This decrease represents a percentage reduction of 35.5%. As Table 1 shows, the reduction in the σ_b_ values showed a slightly decreasing tendency with the number of processing cycles, but the ε_b_ values dropped drastically. Since the value of elongation is in the denominator and this was reduced to a greater extent, the overall effect on this ratio on the modulus was higher than stress. Therefore, the values of E_t_ increased with the number of reprocessing cycles, which indicates greater rigidity of the material associated with brittleness [68]. The E_t_ value of the PLA0 piece was 3580 MPa and it increased up to 3900 MPa in the PLA pieces obtained after the fourth reprocessing cycle (PLA4, PLA5, and PLA6). However, the differences attained in the reprocessed PLA pieces were not significant since all the values were within the standard deviation.

One can observe that the mechanical properties remained in relatively similar values up to the third reprocessing cycle. Above the fourth reprocessing cycle, the reduction in ε_b_ reached significantly lower values and, accordingly, the previously mentioned increase in rigidity and brittleness was observed. Furthermore, by comparing the mechanical properties between the PLA0 and PLA1 pieces, one can observe that the first extrusion cycle did not cause significant changes in the material. This observation suggests that its mechanical recycling for up to four cycles is technically feasible.

In addition to the tensile mechanical properties, the influence of PLA reprocessing cycles on hardness and toughness have also been evaluated. Table 2 summarizes the Shore D hardness and impact strength of the neat PLA (PLA0) piece and the PLA pieces with an increasing number of reprocessing cycles. As reported by Qi et al. [69], there is a direct relationship between the E_t_ and hardness values. Therefore, a slight increase in the E_t_ values were observed, but their standard deviation suggests a slight increasing tendency. Accordingly, there were no significant changes in Shore D values. The neat PLA resulted in a Shore D value of 79.6 and all pieces attained with reprocessed PLA showed similar values, which were comprised in the 78–79 range. A slightly low value was observed for the PLA piece corresponding to the fifth reprocessing cycle, that is, 76.8, but this did not follow any tendency and it was relatively similar to all other pieces. As indicated previously, both σ_b_ and ε_b_ values were highly sensitive to the number of reprocessing cycles, while a slight change in the E_t_ value was observed, which is in accordance with the Shore D values provided herein. Thus, this observation suggests that both E_t_ and, in particular, Shore D hardness were not extremely sensitive to the number of reprocessing cycles.

Another property that is highly sensitive to the reprocessing cycles is toughness. This mechanical property was evaluated by assessing the impact strength using the Charpy pendulum. Impact strength is directly related to mechanical resistant properties (σ_b_) and mechanical ductile properties (ε_b_). It represents the energy a material with a particular geometry can absorb during deformation and failure in impact conditions. As summarized in Table 2, the values obtained by the Charpy impact showed a remarkable decrease with increasing reprocessing cycles. The neat PLA (PLA0) piece presented the highest impact-absorbed energy with an impact strength of around 57 kJ·m^−2^. This value remained virtually unchanged with the first reprocessing cycle, which gives support to the previous conclusions regarding tensile strength and elongation at break. Nevertheless, over the second and the third reprocessing cycle, an evident reduction in energy absorption capacity was attained. Moreover, similar to other mechanical properties, the highest loss of toughness was noticed above the fourth reprocessing cycle (PLA4), reaching values down to 35.5 kJ·m^−2^, which represents a percentage reduction of more than 50%. This behavior has been previously observed in PLA systems with cellulose fibers [70]. These results were already expected by seeing the tensile properties gathered in Table 1, since the impact strength is related to the σ_b_ and ε_b_ values, and both tensile parameters decreased when increasing the reprocessing cycles. This progressive decrease in energy absorption capacity can be linked with the degradation process on PLA with increasing reprocessing cycles [70,71]. In each of these cycles, chain scission occurred, which resulted in PLA molecules with shorter chain lengths. These shorter chains could, however, offer improved mobility and, subsequently, they are considered to be more readily to pack in an ordered structure and increase the degree of crystallinity. This can potentially yield an increase in the degree of crystallinity and, thus, it can result in a more brittle material with a parallel reduction in the deformation capacity by which its energy absorption capacity is reduced [68].

The mechanical properties are entirely related to structural changes during reprocessing cycles. Since impact is one of the most sensitive property to increasing reprocessing cycles, the FESEM study was carried out on the fracture surfaces of the injection-molded pieces after the impact tests. The FESEM micrographs are gathered in Figure 1. Typically, PLA is a brittle polymer and shows a brittle fracture with low (or a lack of) deformation. The reprocessing cycles make this embrittlement more pronounced. Figure 1a shows the fracture surface of a neat PLA (PLA0) piece. It shows a relatively smooth surface. The microcracks were low in height and the round edges observed in the fracture were indicative of some energy absorption during impact. After one reprocessing cycle for the PLA1 piece, the fracture surface morphology was similar to that of PLA0, with a slight increase in roughness (Figure 1b). However, after two reprocessing cycles, more marked crack fronts were observed (Figure 1c) with parallel formations during the growth and propagation of the crack in the fracture process and also with more marked steps. This new morphology induced after reprocessing was indicative of an embrittlement process, which results in less energy absorption during impact conditions. By increasing the number of reprocessing cycles from three to six (Figure 1d–g), the same morphology was attained, which is indicative of the same phenomenon, but it became more evident with the increase in the reprocessing cycle number. There was also a marked increase in surface roughness due to the larger size of the cracks. These cracks were responsible for the final failure of the PLA pieces. These morphologies are in total accordance with the mechanical properties described above and corroborate the lower energy absorption capacity by impact due to degradation of PLA after each reprocessing cycle [71,72].

### 3.2. Influence of the Reprocessing Cycles on the Melt Fluidity, Thermal Properties, and Visual Aspect of PLA

As described above, aliphatic polyesters are particularly sensitive to hydrolytic degradation processes, including chain scission by the effect of water/moisture on the ester groups. This process is shown for PLA in Figure 2 [5,6,22]. It is more pronounced with the effect of temperature. Therefore, reprocessing cycles make PLA even more sensitive to moisture.

This fragmentation results in shorter chains and, evidently, these short chains possess increased mobility, which gives increased flow with temperature. In this sense, a direct measure of a material degradation after reprocessing cycles is the viscosity or the MFI, as proposed by Tochacek et al. [59]. It is true that the MFI does not take into account the variation with the shear rate and temperature as other tests can provide, such as oscillatory rheometry, but MFI is a simple test that requires low-cost equipment and can potentially give detailed information about the degradation a polymer material has undergone [73,74]. Figure 3 plots the evolution of the MFI values with an increasing number of reprocessing cycles. One can observe that, prior to the fourth reprocessing cycle, the percentage increase of MFI ranged between 10% and 20%. However, from the fifth reprocessing cycle, the MFI values increased by 40% when compared to the neat PLA (PLA0) and almost doubled in the sixth reprocessing cycle. These results are consistent with those described above concerning mechanical properties and corroborate that the first reprocessing cycle was not critical in PLA degradation, whereas the second and third cycles induced a slight influence. Nevertheless, above the fourth reprocessing cycle, the effect of degradation was clearly detectable in both mechanical properties and melt fluidity [53,58,68].

As indicated previously, degradation of PLA involves chain scission due to hydrolysis that is accentuated by the heating cycles. In this regard, DSC is a very helpful tool in identifying the main thermal transitions of PLA when subjected to different reprocessing cycles such as *T_g_*, *T_cc_*, and T_m_, among others. It was possible to calculate the degree of crystallinity from the values of the enthalpies corresponding to the melt and cold crystallization processes. Figure 4 shows a comparative plot of the DSC thermograms corresponding to PLA subjected to different reprocessing cycles. Concerning the neat PLA (PLA0) piece, the *T_g_* value was around 60 °C and it was identified by a step-change in the baseline. The cold crystallization process was not detected in the neat PLA and, regarding the melting process, the peak was around 150 °C. However, the intensity of the melting peak was relatively low, which indicates that PLA presented low crystallinity. The DSC thermogram corresponding to the PLA piece produced with the first reprocessing material did not differ from the previous one and, again, the *T_g_* was located at about 60 °C. The melting process (with a T_m_ value close to 150 °C) showed a low intensity, which was almost identical to the neat PLA.

After the second reprocessing cycle, major thermal changes were clearly observable by DSC. The *T_g_* was identified in the same temperature range, of 60 °C, but a cold crystallization peak with a maximum at 120 °C was seen. The melt peak process, located at 150 °C, dramatically increased its intensity compared with that of PLA0 and PLA1. This indicates that the degree of crystallinity of PLA achieved after two reprocessing cycles was remarkably superior and this is directly related to chain scission mentioned above. The newly formed short length PLA chains had more mobility and were more ready to pack in an ordered structure, which leads to increased crystallinity. As the number of reprocessing cycles increased, both the cold crystallization and the melting peaks became more intense [58]. These results agree with the previous MFI values, which suggested an increase of chain mobility with the reprocessing cycles. Table 3 summarizes the most relevant thermal parameters obtained from the second DSC heating of the PLA pieces subjected to different reprocessing cycles.

The values of χ_c_max_ were estimated on the melting enthalpy showing the maximum crystallinity that the material can reach, regardless of the cooling rate. As can be seen in Table 3, neat PLA showed a low crystallinity with a value of χ_c_max_ around 4.3%. Therefore, the neat PLA (PLA0) piece was characterized by a small degree of crystallinity, as suggested by its characteristic DSC thermogram shown in Figure 4. After the first reprocessing cycle, the maximum χ*_c_* remained nearly constant, showing a value of 3%. Nevertheless, after the second reprocessing cycle, the DSC thermogram suggested significant changes by the appearance of the cold crystallization process and an increase in the melting peak process. It also led to a χ_c_max_ value of 17.8%, which represents a dramatic increase in the total crystallinity of PLA, achieved after two reprocessing cycles. As the number of reprocessing cycles increased, the degree of crystallinity also increased to values of almost 30% after six cycles. These results support the previous assumptions related to the melt fluidity behavior and are also in accordance with mechanical properties. Furthermore, the *T_g_* value showed a slight increase after the third reprocessing cycle with some oscillations in the results. This small increase can be explained by the fact that, with the increase in crystallinity, the chain mobility in the amorphous phase was also more restricted due to the presence of the formed crystallites. Regarding the cold crystallization process, the characteristic peaks of *T_cc_* also shifted to lower values due to increased chain mobility after fragmentation. Concerning the T_m_ values, it remained around 150 °C in virtually all materials, which indicates that there were no significant changes in the crystal morphology and shape [75].

In addition to the study of thermal transitions by DSC, a preliminary study of the reprocessing cycles on thermal degradation at high temperatures was carried out by TGA. This technique allows studying the behavior of polymer materials during thermal decomposition. Comparative TGA thermograms of the neat PLA piece and the PLA pieces obtained after different reprocessing cycles are displayed in Figure 5. As expected, all samples (from PLA0 to PLA6) showed a one-step degradation process [76], as one can see in Figure 5a. It is worthy to note that several works have suggested that PLA decomposes in a two-step process when the thermogravimetric analysis is carried out under oxygen atmosphere [77], but it does not have any influence on this analysis since this induces thermo-oxidative degradation.

TGA thermograms suggested very slight changes in the thermal degradation parameters since the curves overlapped. Table 4 summarizes the data of both *T_5%_* and *T_deg_* obtained from thermogravimetric curves (Figure 5a) and first derivate curves (Figure 5b), respectively. PLA0 presented an T_5%_ value of 332.6 °C and a T_deg_ value of 380.1 °C, which are typical degradation values of PLA. It is noticeable that, after two reprocessing cycles, the value of T_5%_ only decreased by 6 °C. Nevertheless, the T_deg_ value remained almost invariable. This indicates that the reprocessing cycles do not have a significant effect on thermal degradation at elevated temperatures. It is noteworthy to mention that, after the second reprocessing cycle, *T_5%_* and *T_deg_* remained nearly constant.

The appearance of the pieces of PLA obtained by injection molding after the different reprocessing cycles are gathered in Figure 6. It is noticeable that, as the number of reprocessing cycles increased, there was a clear tendency to yellowing [78,79]. Table 5 summarizes the color coordinates (*L*a*b**), the color variation measured by Δ*E^*^_ab_* with respect to the neat PLA without any reprocessing cycle (PLA0), and the yellowness index (YI) as indicated by the ASTM E313 standard (D65/10°) for the PLA pieces. Δ*E^*^_ab_* represents the color differences in each coordinate while the YI describes the change in color from white toward yellow.

A simple naked eye observation of Figure 6 shows clear evidence of the reprocessing cycles’ effect on the yellowing and/or change in color of the PLA pieces after the different number of reprocessing cycles. As expected, the *L** value decreased and, the color coordinate *b** (blue to yellow) increased remarkably from 10.75 (PLA0) up to values greater than 20 after the fourth reprocessing cycle. It was also possible to find a clear tendency in the evolution of Δ*E^*^_ab_*, taking the color of the PLA0 piece as the reference, with the reprocessing cycles. A single reprocessing already produced a noticeable difference in color since the Δ*E^*^_ab_* value was 3.6 (ΔE*_ab_  ≥  3.5 and < 5), whereas the YI slightly increased from 42.4 (PLA0) to 48.5 (PLA1). After the second reprocessing cycle, the values of Δ*E^*^_ab_* increased to 11.5, which resulted in samples in which an observer can already notice different colors (ΔE*_ab_  ≥  5), and the YI value suffered an important increase up to 54.2. Furthermore, the color variation as well as the yellowness index columns clearly showed an increasing tendency with reprocessing, as expected due to degradation. This behavior has been previously reported by Carrasco et al. [80].

### 3.3. Influence of the Reprocessing Cycles on the Thermomechanical Properties of PLA

TMA allows for studying the dimensional stability of materials subjected to a temperature program. A preliminary study was carried out to assess the possible effect of degradation due to reprocessing cycles on the CLTE values. This parameter is obtained by calculating the slope of the dimensional change as a function of the temperature, which is plotted in Figure 7. One can observe different zones in the TMA curves. From 20 °C to 50–60 °C, there was a linear correlation between the change in dimensions and temperature. Thereafter, a remarkable change in the slope was observed, which is attributable to the glass transition region. Above *T_g_*, the PLA pieces became more plastic and, subsequently, the change in dimensions increased dramatically. This technique is more sensitive to the cold crystallization process since it induces a contraction and the generation of a more packed structure that leads to a decrease in the slope. As can be seen in plots, *T_cc_* was identified as a peak located at approximately 90 °C. Above this temperature, the cold crystallization was completed and the slope remained constant again. To prevent data variation, the CLTE values were evaluated below *T_g_* and above *T_cc_*, where the material became more stable. As can be seen in the graph, the PLA0 pieces showed higher thermal expansion compared to the PLA pieces produced with reprocessed material and this can be related to the previously mentioned increase in crystallinity.

Table 6 gathers the data corresponding to the CLTE values of the neat PLA piece and the PLA pieces obtained after several reprocessing cycles. The neat PLA (PLA0) showed a CLTE value of 106.7 μm·m^−1^·K^−1^. It can be seen that, apparently, the number of reprocessing cycles did not have any influence on the dimensional stability of the PLA pieces since, even after having undergone four reprocessing cycles, the CLTE values remained nearly constant. Although this coefficient generally decreases with reprocessing since it induces crystallization results in rigid materials, this also has a direct impact on a decrease in dilation with temperature, as suggested by Garancher et al. [81]. Furthermore, the CLTE calculated above *T_g_* showed that the PLA pieces softened, which is reflected in higher values of CLTE. These values presented a slight decrease from 198.2 μm·m^−1^·K^−1^, for the PLA0 piece, to 196.24 μm·m^−1^·K^−1^, for the PLA piece after four reprocessing cycles. However, the change was almost negligible. The *T_g_* values were also calculated by the TMA technique as the first change in the slope and one can observe that they remained almost constant. These results are in accordance with DSC results shown above. Nevertheless, the apparent increase in crystallinity observed by DSC was not detected at the macroscopic scale by TMA characterization.

Lastly, DMTA was used to evaluate the effect of the reprocessing cycles on the dynamic properties. In particular, in Figure 8, the evolution of *G’* and *tan δ* were analyzed as a function of the increasing temperature. Figure 8a shows the evolution of *G’* with temperature. Below 55 °C, *G’* remained almost invariable, but, in the temperature range, it comprised between 55 °C and 67 °C. A dramatic decrease in *G’* occurred. This is related to the glass transition region in which *G*’ decreased by almost three orders of magnitude. Then *G’* stabilized and, between 80–95 °C, it increased again. This indicates more elastic behavior that is attributed to an increase in crystallinity and, subsequently, it is reveals the cold crystallization process. Therefore, it was possible to confirm that DMTA is much more sensitive to detect the cold crystallization than DSC, as mentioned previously. The most relevant information that *G’* evolution can provide is the slight shift of the curves toward higher temperatures with the increasing reprocessing cycles, which is representative of a slight increase in *T_g_*. This can be seen in Figure 8b, which shows the evolution of *tan δ* with temperature for the PLA pieces subjected to different reprocessing cycles. Although there are several methods to assess *T_g_* by DMTA, the most used one is related to the peak maximum of *tan δ*. By using this method, the T_g_ value of PLA0 was close to 62.9 °C while it is slightly increased after reprocessing up to 64.6 °C. These results are in accordance with those observed by DSC analysis, which suggested a slight increase in *T_g_* with the increasing reprocessing cycles. Despite DMTA and TMA being more sensitive to the cold crystallization and its effects on the mechanical and dimensional properties, it is worthy to note that DSC provided not only qualitative information about the increase in crystallinity but also quantitative information.

## 4. Conclusions

This work describes the influence of the reprocessing cycles by extruding the properties of PLA pieces prepared by injection molding. Mechanical ductility is very sensitive to the degradation processes associated with melt reprocessing. In fact, although PLA is intrinsically a fragile polymer, it is further weakened by the application of subsequent reprocessing cycles with ε_b_ values ranging from 9%, for virgin PLA, to values of about 6% for PLA subjected to six reprocessing cycles. Consequently, the E_t_ of the neat PLA piece, which was close to 3.6 GPa, slightly increased up to 3.7–3.9 GPa with the increasing number of reprocessing cycles since ε_b_ decreased more significantly than σ_b_. The Shore D hardness remained almost invariable with values between 76–79, which confirms that hardness is not highly sensitive to the degradation produced after reprocessing. One of the most sensitive property of PLA to the number of reprocessing cycles was the impact strength, which changed from 57.8 kJ·m^−2^, for the neat PLA piece, down to values of 30 kJ·m^−2^ after six reprocessing cycles. This observation confirmed the negative effect of reprocessing on toughness.

The measurement of MFI proved to be a simple tool to quantify degradation. PLA is highly sensitive to the hydrolysis of ester groups and this phenomenon produces chain scission, which leads to shorter chains with improved mobility. Thus, reprocessing led to a noticeable increase in the MFI values from 10 g/10 min up to approximately 20 g/10 min after six reprocessing cycles. DSC revealed a noticeable increase in the maximum crystallinity from values of χ_c_max_ of 4%, for the neat PLA piece, up to values close to 30% for the PLA pieces subjected to six reprocessing cycles. This is clear evidence of degradation since the shorter PLA chains were more readily to pack in an ordered structure, which gives rise to a higher degree of crystallinity. Changes in *T_g_* were not remarkable. However, a slight increase of 2 °C in *T_g_* was detected by all thermal analysis techniques used in this study. The degradation of the material was particularly identified with the yellowing experienced by the PLA pieces with an increasing number of reprocessing cycles. The YI value of the neat PLA piece was close to 42.4 and it increased up to >70 after five reprocessing cycles. The color coordinates indicated an evident loss of transparency, that is, lower *L** values, and a shift toward yellow by the observed increase of the *b** coordinate. The latter color coordinate changed from 10.75 up to values close to 22 after five reprocessing cycles. Therefore, it shows clear evidence of the effect of reprocessing cycles on the color of the PLA pieces. As a general conclusion, this study has revealed the sensitivity of PLA to reprocessing at the level of mechanical, thermal, thermomechanical, and morphological properties. Globally, one can consider that a low number of reprocessing cycles did not affect, to a great extent, the properties of PLA. In particular, low degradation was found between 1 and 3 reprocessing cycles, whereas the main losses were attained when PLA was subjected to more than four reprocessing cycles. The obtained results suggest that PLA, which is a fully bio-based polyester, can be mechanically recyclable for up to three cycles of extrusion and subsequent injection molding and, therefore, it contributes to the development of bioplastic articles that successfully meet with the principles of the circular economy.

## Figures and Tables

**Figure 1 polymers-11-01908-f001:**
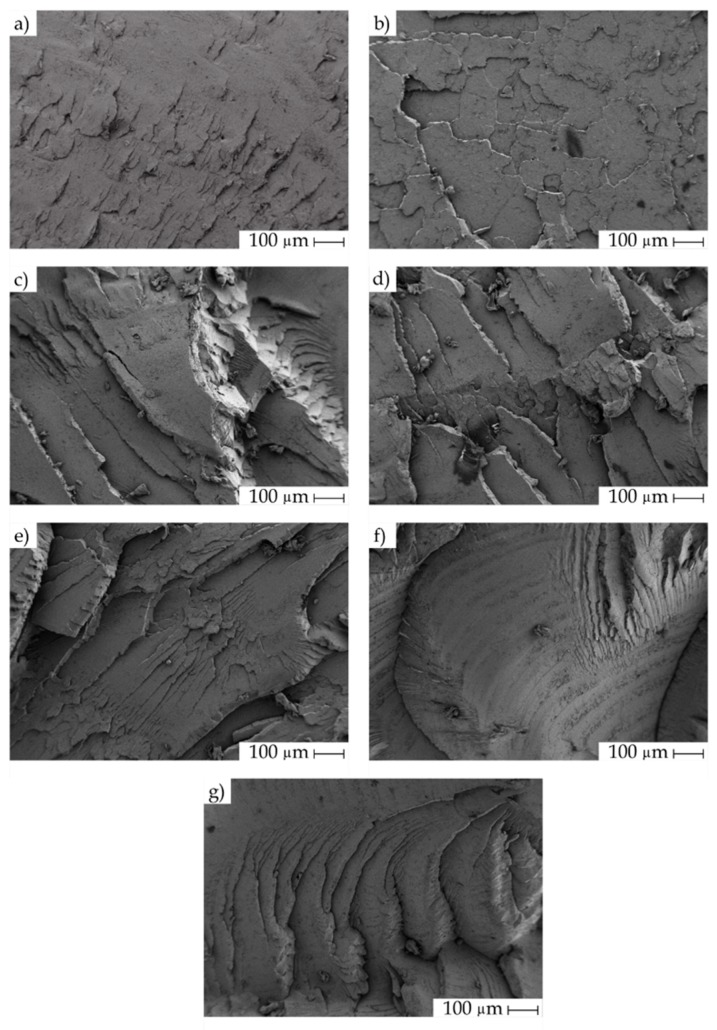
Field emission scanning electron microscopy (FESEM) images of the fracture surfaces of the injection-molded polylactide (PLA) pieces subjected to different reprocessing cycles: (**a**) PLA0, (**b**) PLA1, (**c**) PLA2, (**d**) PLA3, (**e**) PLA4, (**f**) PLA5, and (**g**) PLA6. Images were taken at 1000× and scale markers are of 100 µm.

**Figure 2 polymers-11-01908-f002:**
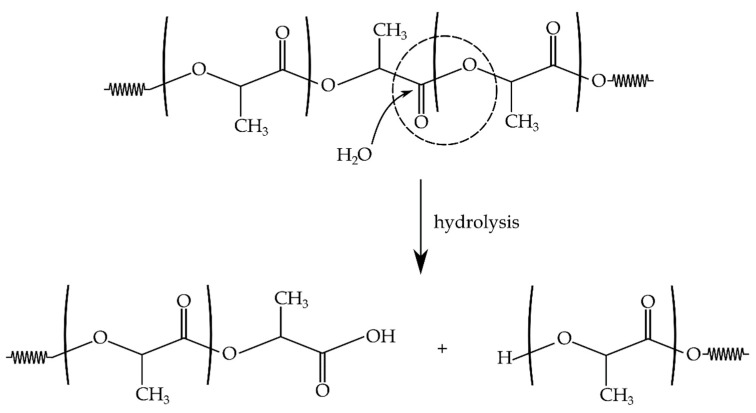
Schematic representation of the hydrolysis fragmentation of ester groups of polylactide (PLA) chains in the presence of water.

**Figure 3 polymers-11-01908-f003:**
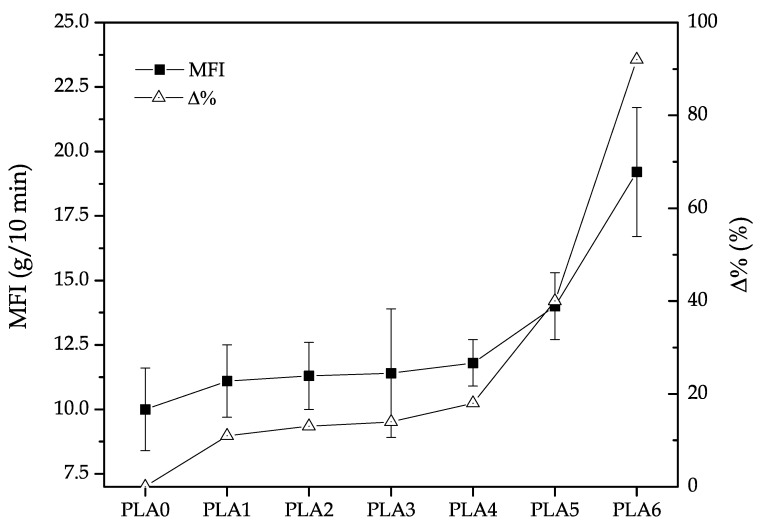
Variation of the melt flow index (MFI) and its percentage increase (Δ%) of polylactide (PLA) subjected to different reprocessing cycles.

**Figure 4 polymers-11-01908-f004:**
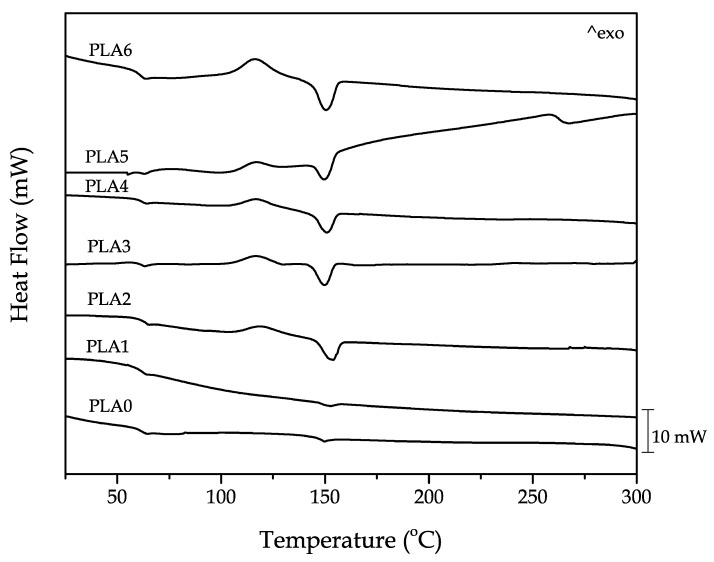
A comparative plot of the differential scanning calorimetry (DSC) thermograms during second heating corresponding to the injection-molded polylactide (PLA) pieces subjected to different reprocessing cycles.

**Figure 5 polymers-11-01908-f005:**
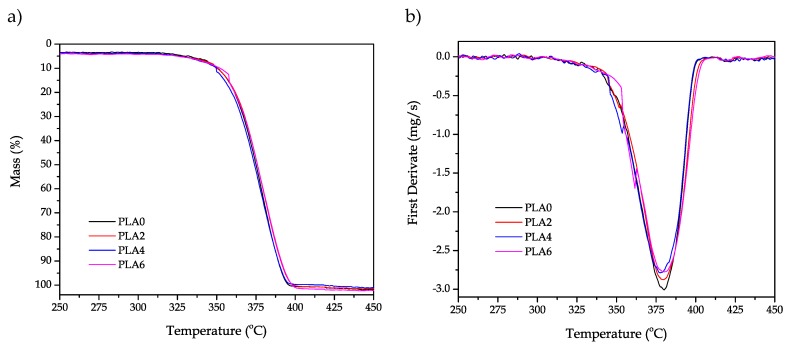
(**a**) Thermogravimetric analysis (TGA) and (**b**) first derivate thermogravimetric (DTG) curves corresponding to the injection-molded polylactide (PLA) pieces subjected to different reprocessing cycles.

**Figure 6 polymers-11-01908-f006:**
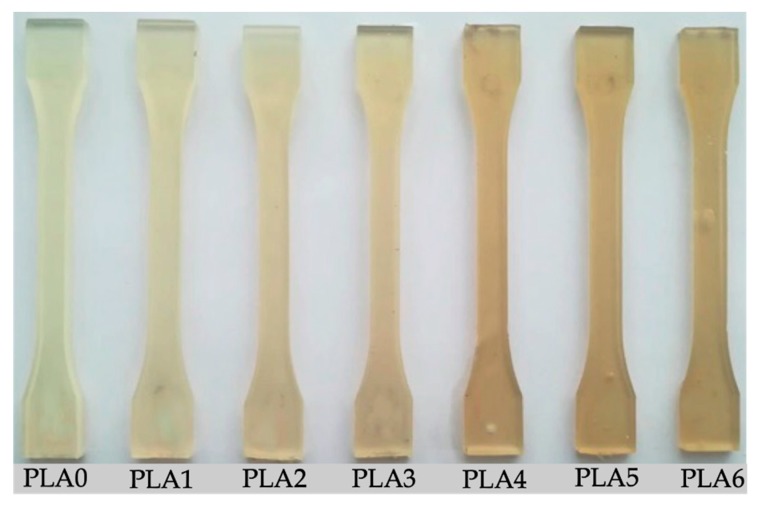
Visual aspect of the injection-molded polylactide (PLA) pieces subjected to different reprocessing cycles.

**Figure 7 polymers-11-01908-f007:**
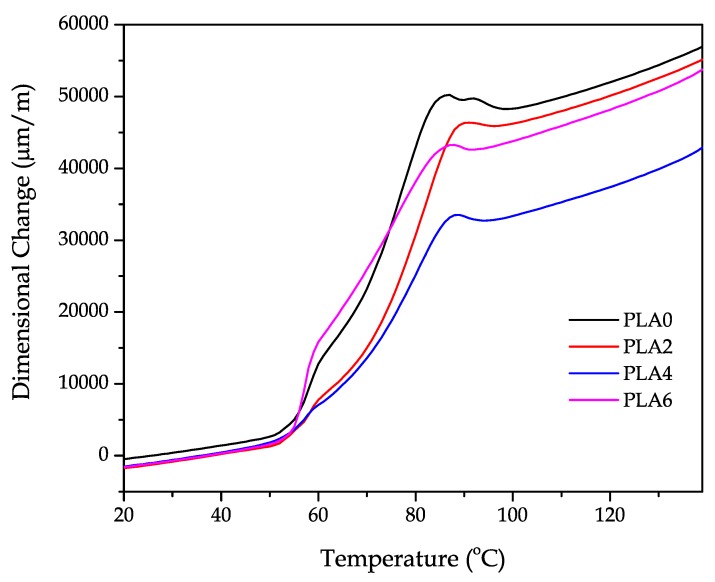
A comparative plot of the dimensional change of the injection-molded polylactide (PLA) pieces subjected to different reprocessing cycles.

**Figure 8 polymers-11-01908-f008:**
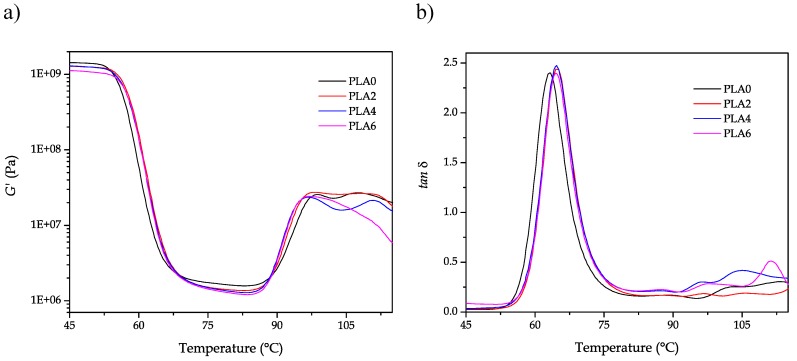
Evolution as a function of temperature of the (**a**) storage modulus (*G’*) and (**b**) dynamic damping factor (*tan δ*) of the injection-molded polylactide (PLA) pieces subjected to different reprocessing cycles.

**Table 1 polymers-11-01908-t001:** Mechanical properties of the injection-molded polylactide (PLA) pieces subjected to different reprocessing cycles in terms of: tensile modulus (E_t_), strength at break (σ_b_), and elongation at break (ε_b_).

Piece	*E_t_* (MPa)	*σ_b_* (MPa)	*ε_b_* (%)
PLA0	3580 ± 180 ^a^	54.8 ± 0.7 ^a^	9.73 ± 0.04 ^a^
PLA1	3670 ± 250 ^b^	54.0 ± 0.8 ^a^	8.80 ± 0.47 ^b^
PLA2	3750 ± 225 ^b^	53.8 ± 0.7 ^b^	8.79 ± 0.33 ^b^
PLA3	3880 ± 290 ^b^	53.4 ± 0.4 ^b^	8.49 ± 0.28 ^b^
PLA4	3900 ± 395 ^b^	52.1 ± 1.0 ^c^	7.27 ± 0.91 ^c^
PLA5	3870 ± 302 ^b^	51.5 ± 1.3 ^c^	6.28 ± 0.21 ^d^
PLA6	3770 ± 245 ^b^	51.2 ± 0.9 ^c^	6.28 ± 0.11 ^d^

^a–d^ Different letters in the same column indicate a significant difference among the samples (*p* < 0.05).

**Table 2 polymers-11-01908-t002:** Mechanical properties of the injection-molded polylactide (PLA) pieces subjected to different reprocessing cycles in terms of D Shore hardness and impact strength.

Piece	Shore D Hardness	Impact Strength (kJ·m^−2^)
PLA0	79.6 ± 3.2 ^a^	57.8 ± 3.7 ^a^
PLA1	79.3 ± 3.3 ^a^	56.3 ± 3.6 ^a^
PLA2	78.1 ± 1.9 ^a^	47.0 ± 2.8 ^b^
PLA3	78.7 ± 2.4 ^a^	41.6 ± 0.6 ^c^
PLA4	78.3 ± 1.3 ^a^	35.5 ± 4.7 ^d^
PLA5	76.8 ± 1.6 ^b^	32.8 ± 2.4 ^d,e^
PLA6	79.1 ± 1.8 ^a^	31.1 ± 1.9 ^e^

^a–e^ Different letters in the same column indicate a significant difference among the samples (*p* < 0.05).

**Table 3 polymers-11-01908-t003:** Thermal properties of the injection-molded polylactide (PLA) pieces subjected to different reprocessing cycles in terms of glass transition temperature (*T_g_*), cold crystallization enthalpy (Δ*H_CC_),* cold crystallization temperature (*T_cc_*), melting enthalpy (Δ*H_m_*), melting temperature *(T_m_*), and maximum degree of crystallinity (χ_c_max_).

Piece	*T_g_* (°C)	Δ*H_cc_* (J·g^−1^)	*T_cc_* (°C)	Δ*H_m_* (J·g^−1^)	*T_m_* (°C)	χ_c_max_ (%)
PLA0	61.4 ± 1.1 ^a^	-	-	4.0 ± 0.2 ^a^	149.4 ± 2.1 ^a^	4.3 ± 0.2 ^a^
PLA1	61.9 ± 0.9 ^a^	-	-	2.5 ± 0.4 ^b^	151.5 ± 1.9 ^a^	2.7 ± 0.8 ^b^
PLA2	63.6 ± 2.1 ^a^	6.5 ± 0.5 ^a^	119.2 ± 1.1 ^a^	16.6 ± 0.7 ^c^	153.7 ± 2.5 ^a^	17.8 ± 0.8 ^c^
PLA3	64.4 ± 2.4 ^a^	10.3 ± 0.6 ^b^	117.1 ± 2.1 ^b^	16.9 ± 0.9 ^c^	150.0 ± 2.0 ^a^	18.2 ± 1.0 ^c^
PLA4	61.9 ± 2.0 ^a^	9.0 ± 0.9 ^c^	116.8 ± 1.7 ^b^	18.2 ± 1.1 ^d^	150.6 ± 1.7 ^a^	19.6 ± 1.1 ^c^
PLA5	63.3 ± 1.9 ^a^	11.4 ± 0.8 ^b^	116.8 ± 2.1 ^b^	16.6 ± 0.9 ^c^	150.3 ± 2.3 ^a^	17.8 ± 1.0 ^c^
PLA6	61.1 ± 2.3 ^a^	19.8 ± 1.0 ^d^	116.0 ± 2.4 ^b^	27.4 ± 1.7 ^e^	150.2 ± 1.9 ^a^	29.5 ± 1.4 ^d^

^a–e^ Different letters in the same column indicate a significant difference among the samples (*p* < 0.05).

**Table 4 polymers-11-01908-t004:** Main thermal parameters of the injection-molded polylactide (PLA) pieces subjected to different reprocessing cycles in terms of onset temperature of degradation (T_5%_) and degradation temperature (T_deg_).

Piece	*T_5%_* (°C)	*T_deg_* (°C)
PLA0	332.6 ± 1.1 ^a^	380.1 ± 1.1 ^a^
PLA2	326.7 ± 0.8 ^b^	379.7 ± 0.9 ^a^
PLA4	326.1 ± 0.9 ^b^	378.1 ± 1.0 ^a^
PLA6	324.7 ± 1.0 ^b^	380.2 ± 1.1 ^a^

^a,b^ Different letters in the same column indicate a significant difference among the samples (*p* < 0.05).

**Table 5 polymers-11-01908-t005:** Color parameters (*L*, a*, b*, and ΔE^*^_ab_*) and yellowness index (YI) of the injection-molded polylactide (PLA) pieces subjected to different reprocessing cycles.

Piece	*L**	*a**	*b**	Δ*E^*^_ab_*	YI (D65/10°)
PLA0	73.23 ± 0.8 ^a^	−6.96 ± 0.08 ^a^	10.75 ± 0.15 ^a^	-	42.4 ± 1.8 ^a^
PLA1	74.15 ± 0.7 ^a^	−4.51 ± 0.06 ^b^	13.23 ± 0.11 ^b^	3.6 ± 0.2 ^a^	48.5 ± 1.6 ^b^
PLA2	63.70 ± 0.8 ^b^	−1.01 ± 0.04 ^c^	13.14 ± 0.14 ^b^	11.5 ± 0.5 ^b^	54.2 ± 1.8 ^c^
PLA3	62.31 ± 0.6 ^b^	−0.53 ± 0.02 ^d^	14.53 ± 0.19 ^c^	13.2 ± 0.6 ^c^	57.5 ± 2.0 ^d^
PLA4	65.06 ± 0.7 ^a,b^	−0.70 ± 0.03 ^e^	21.25 ± 0.20 ^d^	14.7 ± 0.7 ^d^	66.9 ± 3.1 ^e^
PLA5	54.56 ± 0.5 ^c^	0.51 ± 0.02 ^d^	21.77 ± 0.18 ^d^	22.9 ± 1.2 ^e^	73.6 ± 3.3 ^f^
PLA6	58.65 ± 0.9 ^b,c^	−2.10 ± 0.08 ^f^	20.59 ± 0.53 ^e^	18.3 ± 1.1 ^f^	66.8 ± 4.1 ^e^

^a–f^ Different letters in the same column indicate a significant difference among the samples (*p* < 0.05).

**Table 6 polymers-11-01908-t006:** Glass transition temperature (*T_g_*) and coefficients of linear thermal expansion (CLTE) below *T_g_* and above cold crystallization temperature (*T_cc_*) of the injection-molded polylactide (PLA) pieces subjected to different reprocessing cycles.

Code	*T_g_* (°C)	CLTE (μm m^−1^ K^−1^) below *T_g_*	CLTE (μm m^−1^ K^−1^) above Cold Crystallization
PLA0	54.2 ± 1.1 ^a^	106.7 ± 4.3 ^b^	198.2 ± 7.2 ^c^
PLA2	54.9 ± 0.9 ^a^	106.2 ± 5.8 ^b^	195.5 ± 8.6 ^c^
PLA4	53.6 ± 1.7 ^a^	106.3 ± 7.9 ^a^	196.2 ± 9.3 ^c^
PLA6	55.0 ± 2.0 ^a^	111.6 ± 6.2 ^b^	218.4 ± 9.7 ^c^

^a–c^ Different letters in the same column indicate a significant difference (*p* < 0.05).

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
