# Peer review of "Study of the Influence of the Reprocessing Cycles on the Final Properties of Polylactide Pieces Obtained by Injection Molding"

_polymers, 2019, doi:10.3390/polym11121908_

Round 1

Reviewer 1 Report

This manuscript is associated with a significant technological problem. It's interesting and important. However, before accepting it, it must be significantly improved.

Major corrections:

The main problem is the lack of statistical analysis. The authors authoritatively declare (e.g. mechanical properties) which differences are "important" and "significant" and which are not. A statistical analysis of the results should be carried out - its absence adversely affects the quality of interpretation of the results, which is clearly visible in many places of the results description. The description of the results and thirr interpretation must be rewritten and adapted to the results of this analysis. It must be done. Please explain why when the E modulus increased the Shore A hardness decreased? This is surprising and unexpected in the dawn of typical and well known relationships of the module with Shore D hardness values. Please see: Qi, H. J., Joyce, K., Boyce, M. C. (2003), Durometer hardness and the stress-strain behavior of elastomeric materials, Rubber Chemistry and Technology, 76(2), pp. 419–435. We have very strong reduction of hardness (cirka 25%). I have problem to understand your color measurements results. In comparison to what it was done? This is missing in the methodology and results. Why there are no standard deviations? How many samples were used for measurements? In comparison to what ΔE values were calculated? You want to measure changes against PLA0, but then you should not have ΔE for PLA0, because what does this change refer to? Something is wrong here. It would be good to give the values ΔL, Δa, Δb. These ΔE values are absurdly large with such color changes as we see in the picture. This must be explained and described in detail. The conclusions must be changed after following the previous guidelines.

Minor corrections:

-“The effect of increased crystallinity on mechanical properties  is evident since the material becomes stiffer (increased tensile modulus).” I thing that it will be better: “The effect of increased crystallinity on mechanical properties  is evident since the material have increased tensile modulus.” – I fact “tensile modulus” is the material property, stiffness is the property of constructions.

- Shore D measurements- “at least 10 measurements were taken in  different areas from different specimens” – please write how many specimens?

- I suggest a simpler title: Study of the influence of the reprocessing cycles on final properties of PLA polymer

-“3. Results” – should be “3. Results and discussion “

- units formating kJ m-2) etc – please correct it

Author Response

Comments and Suggestions for Authors

This manuscript is associated with a significant technological problem. It's interesting and important. However, before accepting it, it must be significantly improved.  

Major corrections: 

The main problem is the lack of statistical analysis. The authors authoritatively declare (e.g. mechanical properties) which differences are "important" and "significant" and which are not. A statistical analysis of the results should be carried out - its absence adversely affects the quality of interpretation of the results, which is clearly visible in many places of the results description. The description of the results and thirr interpretation must be rewritten and adapted to the results of this analysis. It must be done.

ANSWER

As indicated by the Reviewer, we agree that a statistical analysis is strong enough to declare if the differences are important or not. Therefore, we have carried out an analysis of variance (ANOVA) by using STATGRAPHICS Centurion XVI v 16.1.03. In particular, the Fisher’s least significant difference (LSD) has been provided with regard to mechanical and thermal properties. Accordingly, some comments about these results have been corrected. Moreover, the statistical procedure has been defined in the “Experimental” section (2.8. Statistical Analysis).

Please explain why when the E modulus increased the Shore A hardness decreased? This is surprising and unexpected in the dawn of typical and well known relationships of the module with Shore D hardness values. Please see: Qi, H. J., Joyce, K., Boyce, M. C. (2003), Durometer hardness and the stress-strain behavior of elastomeric materials, Rubber Chemistry and Technology, 76(2), pp. 419–435. We have very strong reduction of hardness (cirka 25%).

ANSWER

We agree with this comment as there is a direct relationship between the Shore D values and the tensile modulus. So that, we have done this characterization again od available samples and we have provided the new values in the revised version. These new calculated values are in accordance with the tensile modulus variation. It is worthy to say that tensile modulus (Et) values have been calculated again by using an axial extensometer which gives accurate values of Et. In this case, we have obtained the typical values for a PLA, i.e. close to 3.5 GPa as also indicated in the technical data sheet of PLA 2003D. We have checked accurately all the measurements and all the data in the manuscript. In particular, new tests regarding tensile modulus and Shore D hardness have been carried out and the obtained results are provided in the text and discussed accordingly. In addition, we have checked the impact strength of all the samples by repeating the impact test on PLA samples subjected to different reprocessing cycles and the provided values in the initial version are correct. We have added the reference by Qi et al. to show the direct relationship between the elastic modulus and Shore D hardness.

I have problem to understand your color measurements results. In comparison to what it was done? This is missing in the methodology and results.

ANSWER

As indicated by the reviewer, additional information about calibration of the colorimeter has been provided in the revised version. Moreover, an additional sentence indicating that a white tile with known L*a*b* coordinates has been used as the base color for comparison. This information has been provided in the “Experimental: 2.7 Colour measurement” section.

Why there are no standard deviations? How many samples were used for measurements?

ANSWER

Initially, the standard deviations were not added to the manuscript to avoid too much numbers but as suggested by the reviewer, it is important to provide them as they give additional information (variability) of the measured property. Therefore, standard deviations have been added in Table 5, and the corresponding ANOVA test. Moreover, we have provided information about the number of samples tested to obtain average colour, in the “Experimental: 2.7 Colour measurement” section.

In comparison to what ΔE values were calculated? You want to measure changes against PLA0, but then you should not have ΔE for PLA0, because what does this change refer to? Something is wrong here. It would be good to give the values ΔL, Δa, Δb. These ΔE values are absurdly large with such color changes as we see in the picture. This must be explained and described in detail.

ANSWER

We agree with the reviewer; it makes no sense E for neat PLA and the presented values are absurdly high (they are wrong calculated). We have come back to the colour coordinates and re-calculated the Eab with regard to the neat PLA (PLA0). Then, the colour changes are referred to the initial colour coordinates of neat PLA L*a*b*: 73,23, -6,96, 10,75). The obtained values are more coherent and represent the colour distance between samples with different reprocessing cycles compared to neat PLA colour. To clarify this, additional sentences have been added in the “Experimental: 2.7 Colour measurement” section.

The conclusions must be changed after following the previous guidelines.

ANSWER

As indicated by the reviewer, the conclusions section has been checked and corrected regarding all the previous observations and new corrected data. 

Minor corrections:  

-“The effect of increased crystallinity on mechanical properties  is evident since the material becomes stiffer (increased tensile modulus).” I thing that it will be better: “The effect of increased crystallinity on mechanical properties  is evident since the material have increased tensile modulus.” – I fact “tensile modulus” is the material property, stiffness is the property of constructions.

ANSWER

We agree with the reviewer. The term stiffness can be confusing as it is widely used in constructions while the tensile modulus is the property of the material. Therefore, we have avoided this confusion by changing the term stiffness by the appropriate sentence or term, mainly “tensile modulus”.

- Shore D measurements- “at least 10 measurements were taken in  different areas from different specimens” – please write how many specimens?

ANSWER

As indicated by the reviewer, the number of samples used to measure Shore D hardness has been provided in the “2.3. Characterization of mechanical properties” section.            

- I suggest a simpler title: Study of the influence of the reprocessing cycles on final properties of PLA polymer

ANSWER

The title has been shortened as suggested by the reviewer. In addition, we have revised the “Abstract” section and re-written some sentences to give a better understanding.

-“3. Results” – should be “3. Results and discussion “

ANSWER

As indicated by the reviewer, the “Results” section has been renamed to “Results and Discussion”

- units formating kJ m-2) etc – please correct it

As suggested by the reviewer, we have checked all the units in the manuscript and corrected some mistakes. Moreover, we have carried out an in-depth revision of the English grammar and spelling and all typos have been corrected.

Reviewer 2 Report

The paper proposes to study the effect of reprocessing on the thermal, rheological, and mechanical properties of PLA. The paper reports extensive characterization of the PLA after reprocessing cycles. The paper is complete and very well-written. I recommend the paper to be accepted after minor revisions. Please see below my comments:

Section 2.2: What is the geometry of the parts that you have injection molded? Did you use the same set of process parameters to mold the parts? Section 2.2: How was reprocessing carried out? Did you took the extruded material and fed that back to the extruder? Did you pelletize the extrude material? Please provide more details about the procedure. Section 2.5: Melt Flow Index represents a point of the flow curve for the polymer and is not able to capture phenomena like shear rate and temperature dependence. Did you carry out other rheological measurements? Please provide more discussion to support the use of MFI as the parameter you consider for the investigation.

Author Response

Comments and Suggestions for Authors

The paper proposes to study the effect of reprocessing on the thermal, rheological, and mechanical properties of PLA. The paper reports extensive characterization of the PLA after reprocessing cycles. The paper is complete and very well-written. I recommend the paper to be accepted after minor revisions. Please see below my comments:  

Section 2.2: What is the geometry of the parts that you have injection molded? Did you use the same set of process parameters to mold the parts?

ANSWER

As indicated by the reviewers, additional information about the geometry of the samples has been provided in the revised version. Moreover, an additional sentence indicating the use of the same set of processing parameters has been added in the “2.2. Processing of materials. Reprocessing cycles by extrusion” section, to avoid confusion.

Section 2.2: How was reprocessing carried out? Did you took the extruded material and fed that back to the extruder? Did you pelletize the extrude material? Please provide more details about the procedure.

ANSWER

As suggested by the reviewer and for the sake of clarity, additional sentences have been provided in the “2.2. Processing of materials. Reprocessing cycles by extrusion” section to give a clear idea of the procedure.

Section 2.5: Melt Flow Index represents a point of the flow curve for the polymer and is not able to capture phenomena like shear rate and temperature dependence. Did you carry out other rheological measurements? Please provide more discussion to support the use of MFI as the parameter you consider for the investigation.

ANSWER

It is true what reviewer says about the limited information a simple MFI test can give, in comparison to other tests such as oscillatory plate-plate rheometry, capillary rheometry, or others. Nevertheless, the usefulness of a simple test such as MFI to give accurate information about the degradation is the focus of this technique in this work which allows calculating a PDI (polymer degradation index) which can be obtained in a low-cost equipment. To support this, we have provided new information and secondary literature.

Reviewer 3 Report

The article presents a very important topic of PLA recycling possibility, which in the opinion of many specialists is a more reasonable utilization alternative compared to composting/degradation. Although my overall impression is very good, I have a few comments that I think should be taken into account by the authors.

The first of my comments concerns the value of tensile modulus, which for most samples is only about 1-1.1 GPa, this value can be assigned for polyolefins, while in the case of PLA it usually exceeds 3 GPa. Please check if the measurement of the module is carried out in the right way, because these types of error suggest a wrong methodology of testing. A similar note applies to the results of impact tests, as the values for PLA usually do not exceed 10 kJ/m2.

As long as there are no objections to the quality of the research, their analysis and conclusions, however, in my opinion the research should be supplemented with an application aspect. The tested PLA variety, like many others, is used for the production of films and thermoformed packaging, in this case the evaluation of transparency and haze index may be the key aspect.

In this context, more detailed rheological properties, including melt strength, may also be important, which may decide about the possibility of film extrusion and thermoforming of finished products.

Author Response

Comments and Suggestions for Authors

The article presents a very important topic of PLA recycling possibility, which in the opinion of many specialists is a more reasonable utilization alternative compared to composting/degradation. Although my overall impression is very good, I have a few comments that I think should be taken into account by the authors.

The first of my comments concerns the value of tensile modulus, which for most samples is only about 1-1.1 GPa, this value can be assigned for polyolefins, while in the case of PLA it usually exceeds 3 GPa. Please check if the measurement of the module is carried out in the right way, because these types of error suggest a wrong methodology of testing. A similar note applies to the results of impact tests, as the values for PLA usually do not exceed 10 kJ/m2.

ANSWER

As indicated by the reviewer, the provided values for the tensile modulus were wrong calculated. A conventional tensile test is not accurate enough to get exact tensile modulus. For this reason, we have tested 5 specimens corresponding to each set of reprocessed PLA to get the tensile modulus by using an axial extensometer which allows obtaining accurate values of stress and strain and, consequently, leading to good tensile modulus values. These new values are in total accordance with other values reported in literature with regard to PLA and as indicated by the provide in the technical datasheet of PLA 2003D. In fact, by using the axial extensometer, the obtained tensile modulus has been 3580 MPa. We have provided some supporting literature about the accuracy of tensile modulus using axial extensometers.

Ferri et al. “Poly(lactic acid) formulations with improved toughness by physical blending with thermoplastic starch”, J. Appl. Polym. Sci. 2018, 135, 45751. Ferri et al. “The effect of maleinized linseed oil as biobased plasticizer in poly(lactic acid)‐based formulations”, Polym. Int. 2017, 66, 882-891. Ferri et al. “Plasticizing effect of biobased epoxidized fatty acid esters on mechanical and thermal properties of poly(lactic acid)”, J. Mat. Sci. 2016, 52, 5356-5366.

Concerning to impact tests, we have tested again 5 specimens of neat PLA without any reprocessing cycle and we have obtained almost identical results as those provided in the manuscript. In addition, to ensure the decreasing tendency, we have tested a set of 3 samples for each reprocessed material and obtained similar values as those included in the original version. Samples were unnotched. We have worked with other PLA grades such as 6201D and it has given impact strength values of about 25-27 kJ/m2 (unnotched samples) and values of about 2-4 kJ/m2 (notched samples, V type).

We have carefully checked all mechanical properties and we have measured again the Shore D hardness as we found an inconsistency with the tensile modulus evolution. The decrease in Shore D hardness is almost negligible. Therefore, we have changed the old values to the new correct values.

As long as there are no objections to the quality of the research, their analysis and conclusions, however, in my opinion the research should be supplemented with an application aspect. The tested PLA variety, like many others, is used for the production of films and thermoformed packaging, in this case the evaluation of transparency and haze index may be the key aspect.

ANSWER

We agree with the reviewer. This PLA commercial grade IngeoTM 2003D is suitable for extrusion but it also gives good processability by injection moulding as reported in this work. This widens the potential of this particular grade in other applications different of film. It is important to remark that this PLA grade gives a high impact strength compared with other injection moulding grades such as 6201D (we have several papers with this material) and this could be an interesting feature to manufacture solid parts with improved toughness. In addition, this grade could also be useful as the base polymer for wood plastic composites as it is specially intended for extrusion. That is why we decided to study this material for new potential applications other than film and thermoforming. Some comments about this have been included in the “Conclusions” section.

In this context, more detailed rheological properties, including melt strength, may also be important, which may decide about the possibility of film extrusion and thermoforming of finished products.

ANSWER

In accordance to the previous question, we are currently working on film formation by using this PLA grade, plasticized with different loads of oligomeric lactic acid (OLA). The properties the reviewer suggests are very useful for this purpose but for injection moulded parts, it is not a key parameter. That is why we have not included the melt strength and other rheologic properties.

Round 2

Reviewer 1 Report

The work has been significantly improved and is definitely better than before. The results have been improved and the authors have done a lot of work to increase the value of the manuscript. I have only one minor suggestion to the Authors, but it is not mandatory. It would be good to include in the discussion short sentences, which of the changes in color are visible to the observer based on dE values, because photography is just an illustration and can distort reality. I can base on:

Stencel, R.; Kasperski, J.; Pakieła, W.; Mertas, A.; Bobela, E.; Barszczewska-Rybarek, I.; Chladek, G. Properties of Experimental Dental Composites Containing Antibacterial Silver-Releasing Filler. Materials 2018, 11, 1031.

Mokrzycki, W.S.; Tatol, M. Color difference ΔE: A survey. Mach. Graph. Vis. 2011, 20, 383–411.

Typical observers see the difference in color as follows:

0 < ∆E < 1 - not noticeable the difference,

1 < ∆E < 2 - only experienced observer can notice the difference

2 < ∆E < 3.5 - unexperienced observer notices the difference,

3.5 < ∆E < 5 - clear difference in color is noticed,

5 < ∆E - observer notices different colors.

You can also comment it in more details, because color is an important practical property, and the differences between the adjacent materials were large. Maybe it would be interesting to count additionally dE values between PLA1 and PLA2, PLA2 and PLA3 etc?

Congratulations to the authors and good luck.

Author Response

Reviewer 1

The work has been significantly improved and is definitely better than before. The results have been improved and the authors have done a lot of work to increase the value of the manuscript. I have only one minor suggestion to the Authors, but it is not mandatory. It would be good to include in the discussion short sentences, which of the changes in color are visible to the observer based on dE values, because photography is just an illustration and can distort reality. I can base on:

Stencel, R.; Kasperski, J.; Pakieła, W.; Mertas, A.; Bobela, E.; Barszczewska-Rybarek, I.; Chladek, G. Properties of Experimental Dental Composites Containing Antibacterial Silver-Releasing Filler. Materials 2018, 11, 1031.

Mokrzycki, W.S.; Tatol, M. Color difference ΔE: A survey. Mach. Graph. Vis. 2011, 20, 383–411.

Typical observers see the difference in color as follows:

0 < ∆E < 1 - not noticeable the difference,

1 < ∆E < 2 - only experienced observer can notice the difference

2 < ∆E < 3.5 - unexperienced observer notices the difference,

3.5 < ∆E < 5 - clear difference in color is noticed,

5 < ∆E - observer notices different colors.

You can also comment it in more details, because color is an important practical property, and the differences between the adjacent materials were large. Maybe it would be interesting to count additionally dE values between PLA1 and PLA2, PLA2 and PLA3 etc?

Answer

As reviewer 1 recommends, both comments about the color change and references were added. this was highlighted with green color for easy reading.

Reviewer 3 Report

Most of the comments have been included in the text. Some doubts about the reliability of the measurements have also been clarified. Some inaccuracies regarding the application aspect could still be supplemented, but I hope that they will be the subject of the next article. In my opinion, the article can be considered for publication.

Author Response

Most of the comments have been included in the text. Some doubts about the reliability of the measurements have also been clarified. Some inaccuracies regarding the application aspect could still be supplemented, but I hope that they will be the subject of the next article. In my opinion, the article can be considered for publication.

Answer

As reviewer 3 suggests, the mentioned points will be taken into account in the preparation of the following work.
